# Learning interpretable dynamics of stochastic complex systems from experimental data

Ting-Ting Gao [1,2], Baruch Barzel [3,4] & Gang Yan [1,2]

Complex systems with many interacting nodes are inherently stochastic and best described by stochastic differential equations. Despite increasing observation data, inferring these equations from empirical data remains challenging. Here, we propose the Langevin graph network approach to learn the hidden stochastic differential equations of complex networked systems, outperforming five state-of-the-art methods. We apply our approach to two real systems: bird flock movement and tau pathology diffusion in brains. The inferred equation for bird flocks closely resembles the second-order Vicsek model, providing unprecedented evidence that the Vicsek model captures genuine flocking dynamics. Moreover, our approach uncovers the governing equation for the spread of abnormal tau proteins in mouse brains, enabling early prediction of tau occupation in each brain region and revealing distinct pathology dynamics in mutant mice. By learning interpretable stochastic dynamics of complex systems, our findings open new avenues for downstream applications such as control.

The behaviors of complex systems, ranging from cell migration to pathological protein diffusion, brain activity, human mobility, and bird flocking, exhibit not only nonlinearity but also stochasticity[1–3]. Stochasticity plays a crucial role in enhancing a system's adaptability to rapidly changing environments[4,5], facilitating information processing[6,7], and increasing robustness[8,9]. The emergence of order from disorder has long fascinated scientists, particularly in the context of system dynamics. While the behaviors of complex systems can be observed experimentally, their underlying dynamics remain elusive. Therefore, stochastic differential equations (SDEs) have been widely employed to model such stochastic systems due to their ability to simultaneously describe deterministic evolution and random fluctuations stemming from unresolved degrees of freedom.

However, conventional SDE models used to describe real-world scenarios have certain limitations, such as predefined forms, simplified physics, and assumed parameter values. Fortunately, the increasing availability of empirical data, including network typologies and node activities, provides an opportunity to shift this paradigm. Instead of modeling the dynamics of a complex system using a predefined SDE, it becomes possible to infer the hidden SDE from observational data on system behaviors.

Discovering the governing laws of dynamics from data has become a prominent field of artificial intelligence-empowered scientific exploration[10–13], making significant progress in recent years[14–29]. Numerous data-driven methods have been proposed to identify ordinary differential equations (ODEs) and partial differential equations for single- and few-body nonlinear systems[14–16], as well as ODEs for large networks[17–19]. However, these methods may not effectively address real systems exhibiting stochasticity. Previous efforts to learn stochastic dynamics have primarily focused on predicting a system's future evolution rather than inferring its underlying SDE[20]. Additionally, the majority of previous methods have

[1]MOE Key Laboratory of Advanced Micro-Structured Materials, and School of Physical Science and Engineering, Tongji University, Shanghai, P. R. China. [2]Shanghai Research Institute for Intelligent Autonomous Systems, National Key Laboratory of Autonomous Intelligent Unmanned Systems, MOE Frontiers Science Center for Intelligent Autonomous Systems, and Shanghai Key Laboratory of Intelligent Autonomous Systems, Tongji University, Shanghai, P. R. China. [3]Department of Mathematics, Bar-Ilan University, Ramat-Gan, Israel. [4]Gonda Multidisciplinary Brain Research Center, Bar-Ilan University, Ramat-Gan, Israel. ✉e-mail: gyan@tongji.edu.cn

been validated on simulated systems with known ground-truth dynamics[24,25], and few have demonstrated the ability to infer real stochastic systems with unknown underlying dynamics (with exceptions like[27,28]).

Here, we aim to address a fundamental question: given the observations of network topology and nodes' activity series, how can we infer the coupled SDEs that capture the hidden stochastic dynamics of a complex system? The main contributions of our work are summarized as follows:

1. We propose a method termed the Langevin Graph Network Approach (LaGNA) that incorporates an innovative message-passing mechanism to separate dynamical sources within nodal activity data. This method subsequently infers concise mathematical expressions for each of these dynamic sources by leveraging corresponding neural network modules. Comparative analyses showcase our method's proficiency in effectively unveiling the hidden coupled SDEs of complex networked systems, demonstrating superior performance compared to five state-of-the-art methodologies in the field.

2. We apply our method to natural flocking, an intriguing phenomenon and important research topic in the community of statistical physics and complex systems. From the trajectory data of several flocks our method successfully infers the SDE of real flocking dynamics. The inferred SDE exhibits a remarkable resemblance to

the second-order Vicsek model, providing unprecedented evidence that the Vicsek model is not just a toy model but captures genuine flocking dynamics.

3. We apply our method to the spreading of pathological tau protein in Alzheimer's disease (AD) brains, a frontier problem in neuroscience. From the experimental data of tau pathology in AD mice brains, our method successfully infers a novel SDE that captures the tau diffusion dynamics. The finding not only enables early-stage prediction of the percentage of brain areas that will be affected by tau pathology but also offers novel quantitative insights into the mechanism of tau pathology.

## Results
### Overview of the LaGNA framework
The state evolution of a complex system is often driven by several dynamic sources, including the self-dynamics of each node, the interaction between nodes, and intrinsic stochastic diffusion. In the first stage of LaGNA (Fig. 1a–d), we design a message-passing mechanism guided by the complex network's nontrivial topology. The message-passing mechanism consists of three neural network (NN) modules: self-dynamics simulator $\hat{\mathbf{f}}(\cdot)$, interaction dynamics simulator $\hat{\mathbf{g}}(\cdot)$, and diffusion simulator $\hat{\boldsymbol{\phi}}(\cdot)$, tailored to separate the dynamical sources hidden in nodes' activity data (Fig. 1b,d) and differing from that used in graph neural network[30].

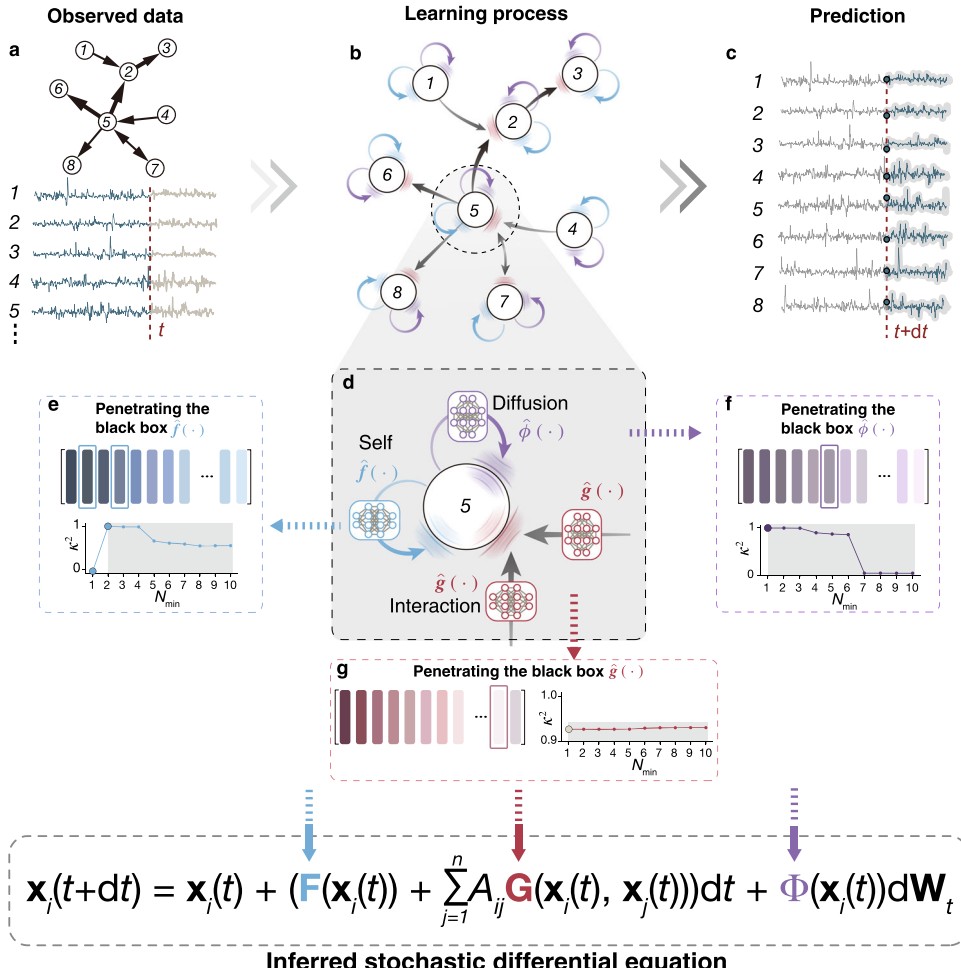

**Fig. 1 | Pipeline of LaGNA. a** Observation data include each node's activity series and the network topology. **b** The stochastic dynamics of a complex networked system consists of three components: self (blue), interaction (gray), and diffusion (purple). **c** Predicted future states of nodes by the well-trained LaGNA. **d** In LaGNA, each node's state evolution is governed by three neural network modules: $\hat{\mathbf{f}}(\cdot)$ for self-dynamics, $\hat{\boldsymbol{\phi}}(\cdot)$ for diffusion, and $\hat{\mathbf{g}}(\cdot)$ for interaction. **e–g** The explicit mathematical expressions of these parts are derived by penetrating the three modules, forming the final stochastic differential equation (SDE), where $N_{min}$ represents the minimal number of elementary terms required.

Each node $i$'s activity at time $t$ is denoted as a $d$-dimensional vector $\mathbf{x}_i(t) \equiv (x_{i,1}(t), x_{i,2}(t), \ldots, x_{i,d}(t))^{\mathsf{T}}$. Given the input of nodes' activities $\mathbf{x}_i(t)$, $i = 1, 2, \ldots, n$ (Fig. 1a), the LaGNA estimates the states at the next time step $\hat{\mathbf{x}}_i(t + \mathrm{d}t)$ (Fig. 1c) using the following equation:

$$\hat{\mathbf{x}}_i(t+\mathrm{d}t) = \mathbf{x}_i(t) + (\hat{\mathbf{f}}(\mathbf{x}_i(t)) + \sum_{j=1}^{n} A_{ij}\,\hat{\mathbf{g}}(\mathbf{x}_i(t), \mathbf{x}_j(t)))\,\mathrm{d}t + \hat{\boldsymbol{\phi}}(\mathbf{x}_i(t))\,\mathrm{d}\mathbf{W}_t. \quad (1)$$

Here, $A_{ij}$ is the adjacency matrix representing the network topology, and $\mathbf{W}_t$ is the $d$-dimensional vector representing the Wiener process (i.e., normally distributed around zero with variance $\mathrm{d}t$). Note that the form of Eq. (1) can describe a wide range of complex dynamical systems[31–33], including epidemic spreading, neuronal dynamics, ecological dynamics, gene regulation, as well as flocking and tau pathology diffusion, as we will show below. The current stage of LaGNA can be viewed as an implicit dynamical system with a large number of trainable parameters: $\boldsymbol{\theta}_{\mathbf{f}}$, $\boldsymbol{\theta}_{\mathbf{g}}$, and $\boldsymbol{\theta}_{\boldsymbol{\phi}}$ representing the self, interaction, and diffusion simulators, respectively. To capture the underlying dynamics of a given complex system, LaGNA's outputs $\hat{\mathbf{x}}_i(t + \mathrm{d}t)$ need to exhibit behavior similar to the true observation $\mathbf{x}_i(t + \mathrm{d}t)$. Due to the intrinsic stochasticity, minimizing the difference between $\hat{\mathbf{x}}_i(t + \mathrm{d}t)$ and $\mathbf{x}_i(t + \mathrm{d}t)$ will result in overfitting. Therefore, we train LaGNA with observation pairs $\mathbf{x}_i(t)$ and $\mathbf{x}_i(t + \mathrm{d}t)$, and obtain its optimal parameters by maximizing instead the expectation:

$$\hat{\boldsymbol{\theta}}_f, \hat{\boldsymbol{\theta}}_g, \hat{\boldsymbol{\theta}}_\phi := \operatorname{argmax}_{\boldsymbol{\theta}_f, \boldsymbol{\theta}_g, \boldsymbol{\theta}_\phi} \mathbb{E}[\log p_{\boldsymbol{\theta}_f, \boldsymbol{\theta}_g, \boldsymbol{\theta}_\phi}(x_i(t+\mathrm{d}t)|x_i(t), \mathrm{d}t)], \quad (2)$$

where $p_{\boldsymbol{\theta}_f, \boldsymbol{\theta}_g, \boldsymbol{\theta}_\phi}$ is the probability density of the normal distribution generated by the model of Fig. 1b with parameters $\boldsymbol{\theta}_f$, $\boldsymbol{\theta}_g$, $\boldsymbol{\theta}_\phi$. Note that Eq. (2) describes the case of $d = 1$; refer to the "Methods" section for situations $d > 1$.

The well-trained LaGNA has the ability to predict future behaviors; however, it currently lacks an explicit equation to describe the underlying dynamics of the system. In the second stage, we aim to unveil the inner workings of the LaGNA black box. The tailored message-passing mechanism (Fig. 1d) has separated the underlying dynamics into three neural network modules, namely $\hat{\mathbf{f}}(\cdot)$, $\hat{\mathbf{g}}(\cdot)$, and $\hat{\boldsymbol{\phi}}(\cdot)$. This decomposition allows us to penetrate each module, deriving explicit expressions for the three parts. Using pre-constructed comprehensive libraries of terms, i.e., $L_F$, $L_G$, and $L_\Phi$ shown in Supplementary Information Section I-B, we identify the optimal combination of terms from the libraries using a modified version of our two-phase approach[17]. Our framework successfully separates and identifies concise mathematical expressions for self-dynamics, interaction dynamics, and intrinsic stochastic diffusion, respectively, which together form the final stochastic differential equation (Fig. 1e,f,g). LaGNA enables the balance of accuracy and complexity of mathematical expressions (see Supplementary Information Section III-C), becoming an interpretable learner for discovering the hidden SDEs of complex networked systems. Further details are described in Methods and Supplementary Information Section I.

## Learning the stochastic dynamics of signed and weighted networks

Signed and weighted networks are prevalent in various biological and physical systems. In neuronal systems, for instance, the synapses between neurons can be either excitatory, enhancing the activity of the receiving neuron, or inhibitory, reducing activity. In physical systems like power grids and traffic networks, link weights play a crucial role in system characterization. The combined effect of heterogeneous links in these networks makes interaction intricate and poses challenges in dynamics inference.

To address the challenge of interaction heterogeneity and validate the effectiveness of our framework, we conduct simulations of a stochastic system with Hindmarsh-Rose (HR) neuronal dynamics on a signed network[34] (refer to Supplementary Information Section II-B.3). In the simulations, we randomly assign half of the nodes as excitatory and the other half inhibitory. The links from excitatory nodes show excitability with $V_{syn} = 2$, while the links from inhibitory nodes show inhibition with $V_{syn} = -1.5$. To infer the hidden HR dynamics, we incorporate the knowledge of link types and utilize two NNs for estimating excitatory and inhibitory interaction dynamics, respectively. It is worth noting that we use only one trial of the nodes' activity sequence. The results in Fig. 2b–d show that our framework accurately estimates the terms of self, diffusion, and, notably, the two types of interactions. The inferred stochastic differential equations (SDEs) successfully reproduce the force field (Fig. 2e) and the stochastic trajectory (Fig. 2f). Additionally, we consider a weighted network $A_{ij}$, where $0 \le A_{ij} \le 1$ and further simulate the network dynamics using stochastic Rössler equations[35] (refer to Supplementary Information Section II-B.2). The results in Fig. 2h–j demonstrate the inability of the stochastic Rössler dynamics on weighted networks. The trajectory generated by the inferred SDEs exhibits similar dynamical characteristics compared to the original trajectory (Fig. 2g, l), and the reproduced force field closely aligns with the true force field (Fig. 2k).

To underscore the significance of our LaGNA method in inferring the stochastic dynamics of complex networked systems, we conduct comparisons between LaGNA and five state-of-the-art methods, namely Modified-SINDy[29], Two-Phase inference[17], SDE-net[25], SVISE[26] and SFI[28], utilizing the stochastic Lorenz networked system (refer to Supplementary Information Section II-B.1). In this model system, each node's state is represented by a three-dimensional vector $\mathbf{x}_i = (x_{i,1}, x_{i,2}, x_{i,3})$, where the intrinsic stochastic diffusion in one dimension (e.g., $x_{i,2}$) can be influenced by another (e.g., $x_{i,3}$). The stochastic intensity is denoted by $1/\sqrt{\gamma}$, with a smaller $\gamma$ indicating higher stochasticity. As shown in Fig. 2m, among the evaluated methods, LaGNA demonstrates significantly reduced errors in inferring the networked SDE, outperforming the other methods by two orders of magnitude. Although SDE-Net, SVISE, and SFI can effectively estimate drift and diffusion, explicit expressions for the networked SDEs remain elusive for the three methods. This challenge arises from the fact that the drift field of networked SDEs encompasses both self-dynamics and interaction effects. In contrast, LaGNA incorporates three specifically designed neural network modules, especially the message-passing module defined on links to capture the interaction dynamics, which together distinguish between self, interaction, and diffusion effects in observation data. This distinction enables accurate learning of networked SDEs from stochastic trajectories, effectively overcoming the limitations of previous "methods" (see Supplementary Information Section V for further comparisons).

Note that there is an interesting method recently introduced for learning macroscopic dynamical descriptions of stochastic dissipative systems[27]. However, the objective of this method differs from ours, as it aims to capture coarse-grained macroscopic behavior rather than the node-level microscopic dynamics required for our exploration of the real complex systems in the following sections. Due to the disparity in objectives and outputs, this method is not included in the comparison tests.

## Learning the dynamics of empirical bird flocks

Collective motions and swarming are fascinating phenomena widely observed in nature[3,36,37], such as bird and fish flocks, cell motions, and bacteria colonies. Understanding how individuals interact when large numbers of individuals move together in groups without colliding, or even when they perform tasks together, like hunting[3,38], has been a topic of widespread discussion. The prevailing consensus on this phenomenon is that the condensation of individuals results from birds being consistent with their neighbors regarding speed, also known as

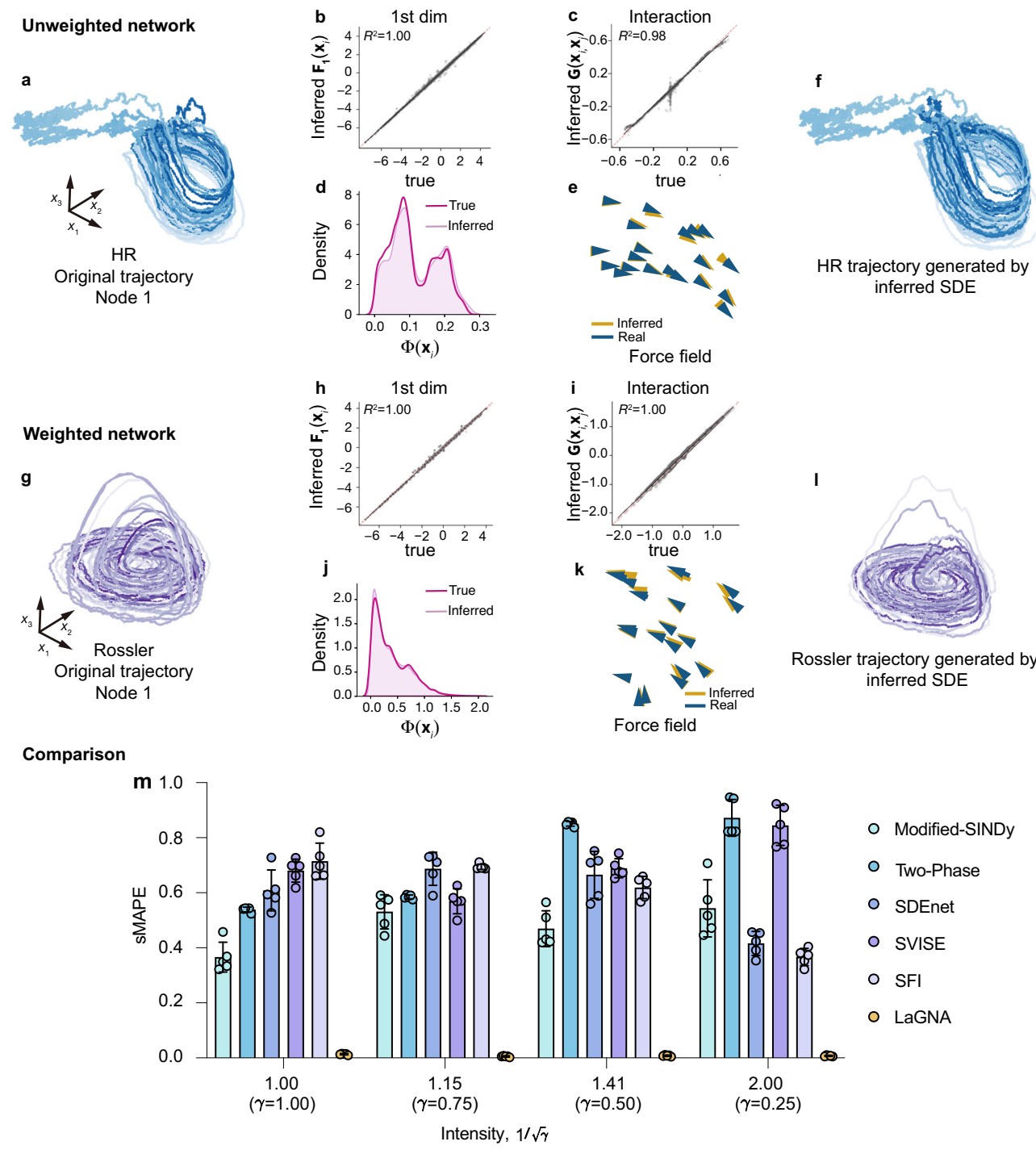

**Fig. 2 | Validating LaGNA on signed and weighted networks. a–f** Inference results of stochastic Hindmarsh-Rose (HR) dynamics on a 20-node signed network, where half of the nodes are excitatory and the other half are inhibitory. **a** Simulated HR stochastic 3-dimensional trajectory of one node. **b–d** Comparisons of the values generated by the inferred SDE and that simulated by the original dynamics model for self (**b**), interaction (**c**), and diffusion (**d**) parts, respectively. **e–f** The force field

and trajectories generated by the inferred SDE. **g–l** Similar to the above, but for the inference results of stochastic Rössler dynamics on a 20-node weighted network, where link weights are randomly drawn from a uniform distribution between zero and one. **m** Inference inaccuracy of previous five methods compared to LaGNA in stochastic Lorenz network dynamics benchmark. The bar chart depicts the sMAPE of five independent runs, along with the standard deviation indicated.

alignment, as well as their tendency to maintain a close distance while avoiding collisions, also known as cohesion[3,39].

To discover the underlying dynamics from the flocking trajectories, we extend the internal architecture of the LaGNA based on the above hypothesis. Specifically, we implement a second-order version by setting up three specialized NNs for simulating the self-propulsion, cohesion, and alignment respectively. We modify the loss function as the summation of negative log-likelihood loss $\mathcal{L}_{nl}$ and three prediction

errors:

$$\mathcal{L} = \beta_1 \mathcal{L}_{nl} + \beta_2 \mathcal{L}_r + \beta_3 \mathcal{L}_v + \beta_4 \mathcal{L}_a, \qquad (3)$$

where $\beta_1, \beta_2, \beta_3, \beta_4$ are hyperparameters balancing the different parts of loss, and $\mathcal{L}_r, \mathcal{L}_v$ and $\mathcal{L}_a$ are the squared error between the predicted and true displacements, velocities and accelerations, respectively. To validate the potency of the extended framework, we generate a 20-bird flocking

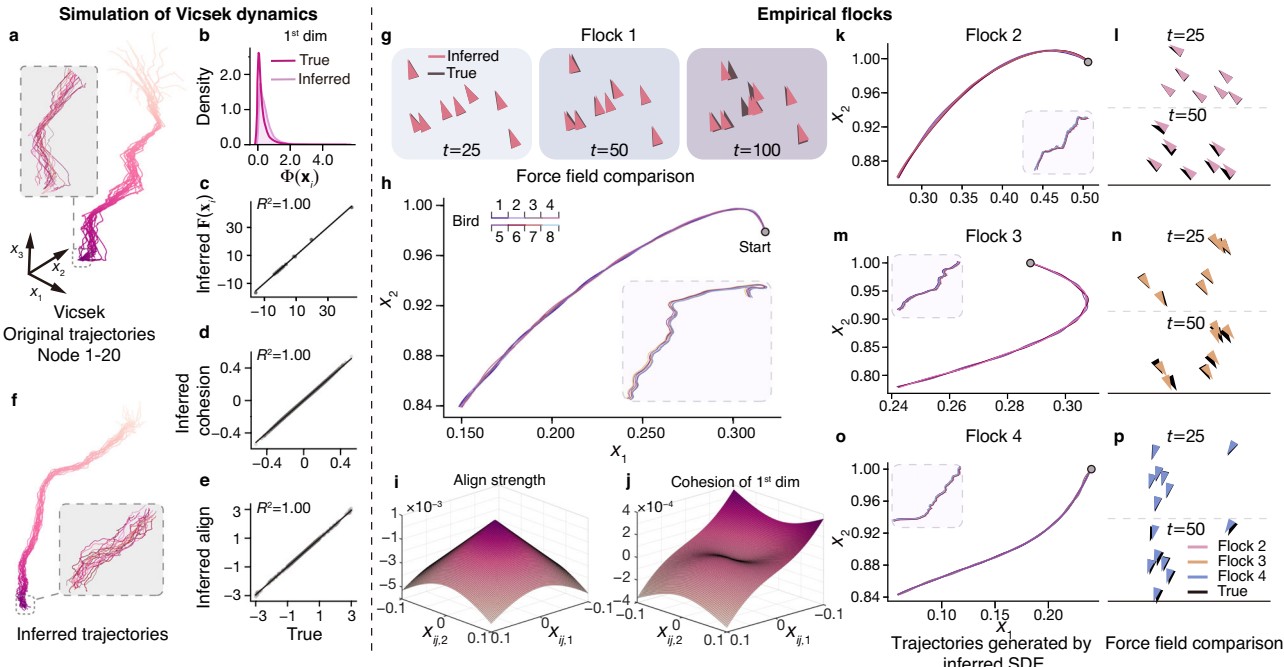

**Fig. 3 | Inferring flocking dynamics from both simulation (a–f) and experimental (g–p) data. a** The 3-dimensional trajectories of 20 birds generated by the 2nd-order Vicsek model. **b–e** The diffusion, self, cohesion, and alignment parts of the first dimension estimated by $\phi(\cdot)$, $f(\cdot)$, $g_a(\cdot)$ and $g_c(\cdot)$, respectively, and their comparison with the true values of the Vicsek model. **f** The flocking trajectories generated by the inferred stochastic Vicsek dynamics. **g–o** Inferring the flocking dynamics from four empirical datasets[40]. **g** Comparison between the force field generated by the inferred SDE and that calculated from the empirical data for 8 pigeons at different time stamps $t = 25, 50, 100$. (**h**) The flocking trajectories generated by the inferred SDE. The subplot is the empirical trajectories from the first flock data. (**i–j**) The visualization of the align and cohesion functions of the inferred SDE. **k–p** The trajectories and force fields generated by the inferred SDE for the second, third, and fourth flock data, respectively.

system with the 3-dimensional Vicsek model. The results show that our framework accurately estimates the self-propulsion, cohesion, and alignment strength of the Vicsek model system, as shown in Fig. 3a–e. The inferred second-order SDEs successfully regenerate the collective behaviors, as detailed in Supplementary Information Section II-B.4.

Furthermore, we apply our extended second-order SDE inference framework to learn the empirical dynamics of bird flocks. The dataset consists of four sets of homing flights, which were collected from homing pigeons equipped with GPS devices[40]. The pigeons were released approximately 15 km away from their home loft, and the GPS devices recorded their location during the return journey at a sampling rate of 0.2 seconds. Because there is less variation in the vertical direction of the bird flocks, here we primarily focus on the movement in the horizontal plane and perform the following data preprocessing steps: spline interpolation with a sampling rate of 0.01 for data augmentation; normalization of the data; extraction of the time period after takeoff and before descent when collective behaviors are most prominent; and alignment of the coordinates. Note that some pigeons exhibited outliers, so their data was removed. In the end, the first flock set contains 8 pigeons, and the other three sets contain 7 pigeons each.

Using the extended framework, we successfully learn the self-propulsion, alignment, and cohesion parts based on one of the four flock datasets. The estimated strengths exhibit a close correspondence with specific scaled functions. Specifically, the alignment strength matches with $\hat{\mathcal{A}} = a_1(\exp(-r_{ij}/3) + a_2) + a_3$, the cohesion strength matches with $\hat{\mathcal{C}} = c_1((r_{ij}/2 - 1)^3/(r_{ij}/2 + 1)^6 + c_2) + c_3$, and the self-propulsion strength is $s_1(|\mathbf{v}_i|^2 + s_2) + s_3$ (refer to Supplementary Information Section III-A). Therefore, the inferred SDE is

$$\dot{\mathbf{v}}_i = s_1\left(|\mathbf{v}_i|^2 + s_2\right) + s_3 + \sum_{j \neq i}\left(\hat{\mathcal{C}}(r_{ij})\mathbf{r}_{ij} + \hat{\mathcal{A}}(r_{ij})\mathbf{v}_{ij}\right) + \hat{\boldsymbol{\epsilon}}\, d\mathbf{W}_t, \quad (4)$$

where $\mathbf{v}_i = \dot{\mathbf{r}}_i$, $\mathbf{r}_{ij} = \mathbf{r}_j - \mathbf{r}_i$, $\mathbf{v}_{ij} = \mathbf{v}_j - \mathbf{v}_i$, $r_{ij} = |\mathbf{r}_{ij}|$, $\mathbf{W}_t \sim \mathcal{N}(0, d\,t)$ representing the Wiener process with mean zero and variance $dt$ with $dt = 0.01$, and $\hat{\boldsymbol{\epsilon}}$ is the estimated intensity of stochasticity. The reproduced force field exhibits consistency with the actual field for a substantial duration across each flock system, and long-term predictions reveal a diverse range of behaviors, as depicted in Fig. 3g–j. To assess the generalizability of the inferred SDE, we employ it to describe three other datasets that were not used in training. We observe that by solely fine-tuning the scaling coefficients without altering the equation form, Eq. (4) is able to effectively capture the underlying dynamic mechanism of the collective behaviors exhibited in these three datasets, as illustrated in Fig. 3k–p. The hyperparameters and coefficients are shown in Supplementary Information Table 5.

The renowned Vicsek model has long served as a staple in flocking dynamics research, often regarded as a simplistic representation. Our finding offers unprecedented evidence that the Vicsek model transcends its toy model status, effectively encapsulating authentic flocking dynamics. Remarkably, Eq. (4) is autonomously inferred from the observation data, devoid of preconceived assumptions about its structure. Consequently, the striking resemblance of the inferred SDE to the second-order Vicsek model[41,42] unveils new perspectives for understanding and modeling the collective behaviors of real flocks.

## Learning the spreading dynamics of tau pathology in mouse brain

Tau proteins play a crucial role in maintaining the stability of axon microtubules, which is essential for the proper functioning of the brain[43]. However, in Alzheimer's disease (AD), misfolded and hyperphosphorylated tau proteins lose their ability to bind microtubules properly, leading to their accumulation as neurofibrillary tangles, a hallmark of the disease[44]. Previous experimental studies have shown that in the early stages of AD, pathological tau spreads from the

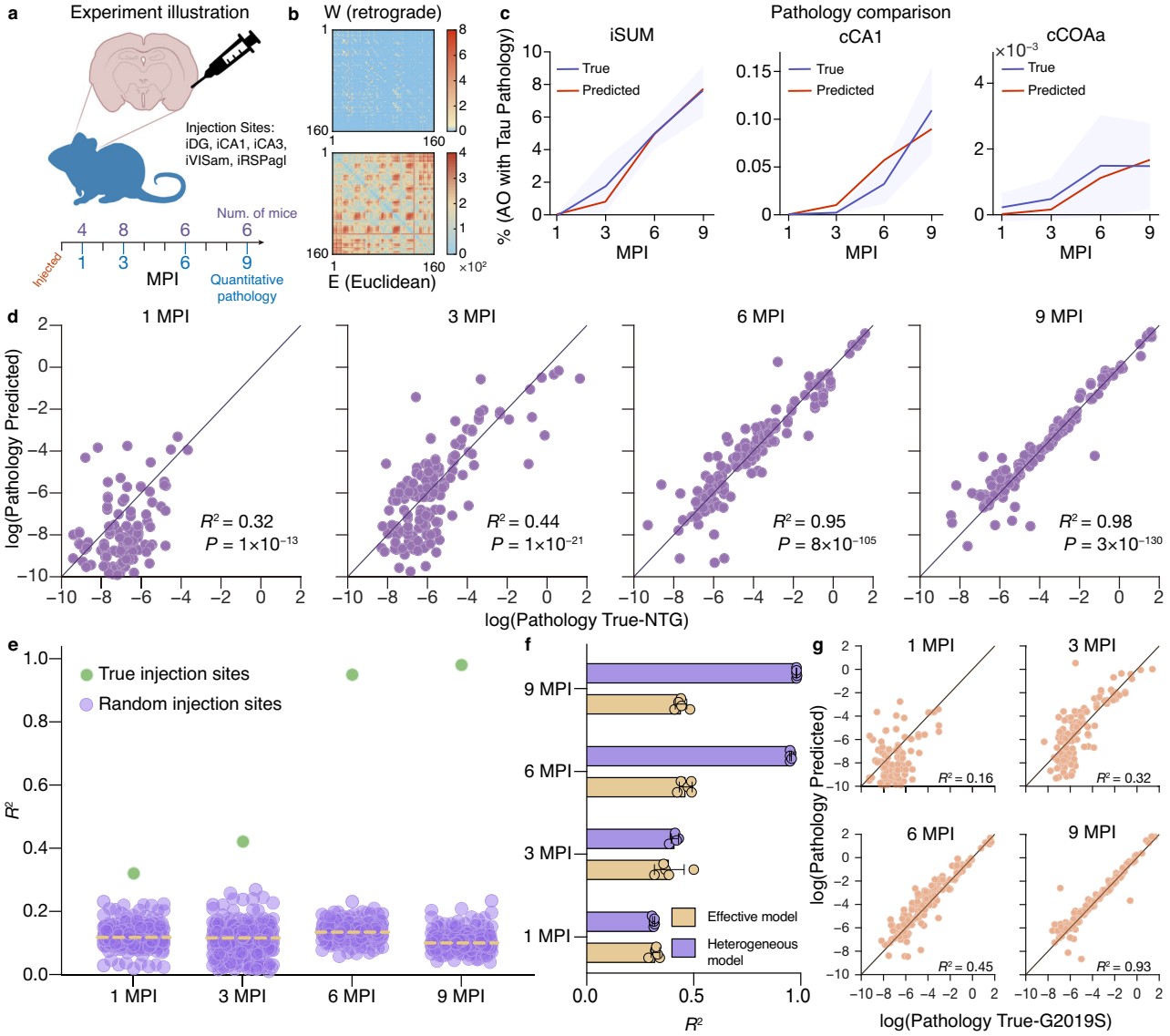

**Fig. 4 | Inferring AD tau pathology diffusion dynamics from experimental data.** **a** Experiment data. Nontransgenic (NTG) mice were injected unilaterally with AD PHF tau in the selected injection sites at 3–4 months of age. Four, eight, six, and six mice were euthanized at 1, 3, 6, and 9 months post-injection (MPI), and the pathology was recorded respectively[48,49]. **b** The anatomical connections in the retrograde direction (upper) and the Euclidean proximal connections (bottom). **c** Comparison between true and predicted area occupied (AO) with tau pathology in site iSUM, cCA1, and cCOAa. The true trajectories represent the average of pathology observed in different mice at 4 different months. The error shades indicate the standard deviation (std). **d** The predictions of log tau pathology from

the inferred diffusion dynamics compared to the true log tau pathology at 1, 3, 6, and 9 MPI, respectively. The square of Pearson correlation $R^2$ and P-values for multiple linear regression tests are noted. **e** The pathology spread patterns predicted from randomly selected 5 injection sites for 500 independent runs (purple dots) v.s. the predicted results starting from the true 5 injection sites (green dots). Dashed lines indicate median values. **f** Comparison between the predictions from the implicit effective model and the explicit heterogeneous diffusion model. Error bars indicate std. **g** The effectiveness of the inferred diffusion equation in the LRRK2$^{G2019S}$ mice pathology data.

transentorhinal cortex to other areas of the limbic and neocortical regions, suggesting a spread along neuroanatomical connections[45]. There is also evidence indicating that tau can be released into the extracellular space, either as free tau or in vesicles such as exosomes[46,47]. As the second empirical system, here we apply our proposed inference framework to the observed tau pathology data to identify the hidden governing equation of the spreading dynamics contributed by both neuroanatomical connections and spatial proximity due to extracellular diffusion.

To obtain the tau spreading data, biologists injected non-transgenic (NTG) mice in five specific injection sites with paired helical filament (PHF) tau extracted from the hippocampus and overlaying cortex. The injected mice were euthanized at different

time points (1, 3, 6, and 9 months after injection) to obtain pseudo-longitudinal data[48,49]. The brain sections of each mouse were then stained to label the percentage of infected areas in different brain regions, as shown in Fig. 4a. We consider the bidirectional diffusion of tau pathology along neuroanatomical connections, with retrograde (from terminals to the cell body) and anterograde (from cell body to terminals) directions[49,50]. We also take into account the influence of geographical distance on diffusion, as shown in Fig. 4b. This leads to a total of $n = 160$ brain regions, with time-dependent percentages of the area occupied by tau pathology denoted as $\mathbf{y}(t)$. We have neuroanatomical and Euclidean weighted adjacency matrices $A$ and $D$, respectively, with the anterograde matrix represented as $A^T$.

By applying our framework, we infer the governing equation of the tau pathology diffusion dynamics as follows:

$$
y_i(t) = \left( b_{0i} x_i(0) + b_{1i} \sum_{j=1}^{n} \tilde{A}_{ij} \frac{x_j(0)}{1+e^{-A_{ij}}} + b_{2i} \sum_{j=1}^{n} \tilde{A}_{ji} \frac{x_j(0)}{1+e^{-A_{ji}}} \right.
$$
$$
\left. + b_{3i} \sum_{j=1}^{n} \tilde{D}_{ij} x_j(0) e^{D_{ij}} \right) \cdot T(t) + \sigma_i(t). \tag{5}
$$

Here, $D_{ij} = 1/\log(E_{ij}^2)$, where matrix $E$ represents the actual Euclidean distance between different regions, and the term $T(t) = c_t + 1.5t$ includes trainable parameter $c_t$ that captures the varying propagation rate over time. The elements in $D$ less than 0.11 are set to zero, meaning that there is no immediate spatial diffusion between two regions that are far apart (refer to Supplementary Information Section III-B). The binary matrix $\tilde{A}$ has elements $\tilde{A}_{ij} = 1$ when $A_{ij} > 0$, and $\tilde{A}_{ij} = 0$ otherwise; The same applies to the binary matrix $\tilde{D}$. In the initial state $x(0)$, only the five sites that were injected with a volume of $1\,\mu g$ of tau are set to a value of 1 and the rest are zero, consistent with previous work[49]. The weights $\mathbf{b}_0$, $\mathbf{b}_1$, $\mathbf{b}_2$ and $\mathbf{b}_3$ are heterogeneous factors assigned to different regions[51]. The term $\boldsymbol{\sigma}$ represents the stochastic noise in the system. It is worth mentioning that the three terms on the right side of the inferred equation correspond to retrograde, anterograde, and spatial diffusion, respectively, which demonstrates the biological interpretability of our inference result.

Due to the stochastic nature of pathology propagation in brains, the pattern emergence occurs following the fluctuating duration of early stages. As shown in Fig. 4c,d, the inferred equation adeptly predicts the tau diffusion at 6 and 9 months post-injection (MPI) given the injection sites. To validate the prediction's specificity to the injection sites, we assess its performance against 500 randomly selected initial sets comprising five regions each. The results affirm that injections with the experimental seed regions yield the highest accuracy across all time points (Fig. 4e). Furthermore, we evaluate a degenerate model that treats each brain region equally. The outcomes demonstrate reduced predictability of the degenerated model (Fig. 4f), underscoring the significant influence of regional heterogeneity on tau pathology diffusion.

Finally, we apply our method to infer the tau diffusion dynamics in mice with LRRK2$^{G2019S}$ mutation. This mutation is the most common cause of familial Parkinson's disease and a common risk factor for idiopathic Parkinson's disease. Mice carrying this mutation exhibit altered tau pathology patterns, but, intriguingly, the inferred equation that accurately delineates tau diffusion in mutant mice (Fig. 4g) shares the same form as Eq. (5). The remarkable distinction lies in the observation that while tau pathology diffusion in NTG mice lacks a directional preference, the diffusion in mutated mice exhibits a pronounced inclination towards the retrograde direction. This preference is quantified by the absolute average value of coefficient $\mathbf{b}_1$ of the retrograde direction ranging from 0.7–0.9, contrasting with $\mathbf{b}_2$ for the anterograde direction, which falls within the range of 0.1–0.3. These results align with a recent experiment[50].

Our discovery offers new insights into tau pathology. Firstly, the inferred Eq. (5) holds promise in generally capturing tau pathology dynamics in brains. Secondly, the results shed light on the significance of spatial diffusion, a factor overlooked in previous studies, indicating its non-negligible impact on tau pathology. Lastly, the delineation of coefficients for retrograde and anterograde diffusion terms underscores the distinct tau pathology dynamics in mutant mice. These findings collectively enhance our understanding of tau pathology mechanisms.

## Discussion

Inferring the governing equations of complex systems from observation data is a crucial direction to automate scientific discovery. Previous studies have primarily focused on benchmarking algorithms on model systems with known ground truths. In contrast, our work delves into two important real-world systems and successfully distills their concealed networked SDEs. This not only showcases the applicability of our approach but also generates novel insights for understanding the mechanisms hidden in empirical flocking and tau pathology diffusion. Importantly, our LaGNA method requires only one trial of nodes' activity sequence and only snapshots rather than continuous time series data, enhancing its flexibility and adaptability to other real scenarios with the aid of inductive bias.

While LaGNA demonstrates superior performance compared to previous state-of-the-art methods and provides valuable insights into real complex systems, it does have limitations that necessitate further attention in future research. Firstly, in some scenarios, the activity time series of certain nodes may be inaccessible. Therefore, it is worth determining the minimal sub-network structure required to unveil the system dynamics[52–54]. Real data from stochastic systems often exhibit a combination of intrinsic stochasticity and extrinsic noise, with the latter arising from measurement errors. Distinguishing between these types of noise poses significant challenges[55–57]. Without prior knowledge of the dominant source of noise, we treat all noises as intrinsic in this work, and LaGNA demonstrates accurate inference when the relative strength of extrinsic noise is below 10%. When extrinsic noise is more pronounced, a preprocessing step of denoising, such as the Kalman-Takens filter[57], enhances inference capability (see Supplementary Information Sections V-B and V-C). However, future efforts are needed to better address extrinsic noises in data.

Secondly, while many real complex networks have been successfully mapped in the past, obtaining the topological data of a network may not always be feasible in certain scenarios. In such cases, there is a need to infer both network topology and system dynamics. Recent commendable efforts have been made to address this challenge[18,58,59], yet they either require activity data from many trials with different initial states[18] or learn for dynamics prediction rather than inference[58,59]. Simultaneously inferring both the dynamical equation and network topology of a large real system using a limited amount of experimentally feasible data remains a challenging task.

Thirdly, while the pre-constructed libraries in the second stage of LaGNA can contain a large number of orthogonal or non-orthogonal elementary function terms, it is still possible that the use of pre-constructed libraries may overlook certain terms. Symbolic regression, an alternative method that does not rely on pre-constructed libraries, faces higher-dimensional challenges. Thus, further efforts are needed to enhance the automation of current methods.

Fourthly, there has been considerable interest in higher-order interactions within complex systems in recent years[59–61]. LaGNA can be extended to accommodate higher-order systems by incorporating additional terms such as $\sum_{j,k} A_{i,j,k} \mathbf{h}(\mathbf{x}_i(t), \mathbf{x}_j(t), \mathbf{x}_k(t))$ into the interaction part of Eq. (1) where $A_{i,j,k}$ represents the third-order network and the function $\mathbf{h}$ denotes the third-order interaction dynamics. Yet this will increase the complexity in identifying an optimal equation, Yet this extension will increase the complexity of identifying an optimal equation, presenting a promising avenue for future efforts to address.

## Methods
### Loss function of LaGNA
Consider a complex networked system whose dynamics are governed by stochastic differential equations (SDEs)

$$
d\mathbf{x}_i(t) = \left( \mathbf{F}(\mathbf{x}_i(t)) + \sum_{j=1}^{n} A_{ij} \mathbf{G}(\mathbf{x}_i(t), \mathbf{x}_j(t)) \right) dt + \mathbf{\Phi}(\mathbf{x}_i(t)) d\mathbf{W}_t. \tag{6}
$$

Here, $\mathbf{x}_i(t)$ represents the $d$-dimensional state of node $i$ at time $t$; $A$ is the adjacency matrix of size $n \times n$, where $A_{ij}$ denotes the influence from node $j$ to $i$; $\mathbf{F} \equiv (F_1(\mathbf{x}_i), F_2(\mathbf{x}_i), \ldots, F_d(\mathbf{x}_i))^{\mathrm{T}}$ and $\mathbf{G} \equiv (G_1(\mathbf{x}_i, \mathbf{x}_j), G_2(\mathbf{x}_i, \mathbf{x}_j), \ldots, G_d(\mathbf{x}_i, \mathbf{x}_j))^{\mathrm{T}}$ are nonlinear functions representing the self and

interaction dynamics, respectively; $\boldsymbol{\Phi}(\mathbf{x}_i(t))$ is the positive-definite diffusion matrix of size $d \times d$, and $\mathbf{W}_t$ is a $d$-dimensional vector representing the Wiener process with mean zero and variance $\mathrm{d}t$[25]. Note that, by choosing different $\mathbf{F}$ and $\mathbf{G}$, Eq. (6) can describe a wide range of systems dynamics[32,33].

For simplicity, let's consider first the case $d = 1$. Given $\mathbf{x}(t)$ and $\mathrm{d}t$, $\mathbf{x}(t + \mathrm{d}t)$ can be considered as points drawn from the normal distribution

$$x_i(t + \mathrm{d}t) \sim \mathcal{N}(\mu_i(t), \sigma_i^2(t)), \tag{7}$$

where $\mu_i(t) = x_i(t) + [F(x_i(t)) + \sum_{j=1}^{n} A_{ij} G(x_i(t), x_j(t))]\mathrm{d}t$, and $\sigma_i^2(t) = \Phi^2(x_i(t))\mathrm{d}t$. To train the network end to end, we use all nodes' states at time $t$, $\mathbf{x}(t)$, as inputs. Based on the network topology $A_{ij}$, we map the information flow from node $j$ to node $i$ using a function $g(x_i(t), x_j(t))$. The estimated information values are then aggregated element-wise for each receiving node over all respective sending nodes. Additionally, we map the self-dynamics of each node $i$ using a function $f(x_i(t))$. The estimated mean and variance of node $i$'s activity distribution can be written as $\tilde{\mu}_i(t) = x_i(t) + [f(x_i(t)) + \sum_{j=1}^{n} A_{ij} g(x_i(t), x_j(t))]\mathrm{d}t$ and $\tilde{\sigma}_i^2(t) = \phi^2(x_i(t))\mathrm{d}t$ respectively. The functions $g, f$ and $\phi$ are determined by trainable parameters $\boldsymbol{\theta}_f$, $\boldsymbol{\theta}_g$ and $\boldsymbol{\theta}_\phi$, respectively.

TO obtain the optimal parameters, we train the model in Fig. 1b by maximizing the likelihood between the true and estimated distributions. Since the true distribution is inaccessible and only the next-step state is available, we instead maximize the following expectations using the maximum likelihood estimates (MLE):

$$\hat{\boldsymbol{\theta}}_f, \hat{\boldsymbol{\theta}}_g, \hat{\boldsymbol{\theta}}_\phi := \arg\max_{\boldsymbol{\theta}_f, \boldsymbol{\theta}_g, \boldsymbol{\theta}_\phi} \mathbb{E}[\log p_{\boldsymbol{\theta}_f, \boldsymbol{\theta}_g, \boldsymbol{\theta}_\phi}(x_i(t + \mathrm{d}t)|x_i(t), \mathrm{d}t)], \tag{8}$$

where $p_{\boldsymbol{\theta}_f, \boldsymbol{\theta}_g, \boldsymbol{\theta}_\phi}$ represents the probability density of the normal distribution generated by the model of Fig. 1b with parameters $\boldsymbol{\theta}_f, \boldsymbol{\theta}_g, \boldsymbol{\theta}_\phi$, i.e.,

$$p(x_i(t+1); \tilde{\mu}_i(t), \tilde{\sigma}_i^2(t)) = \frac{1}{\tilde{\sigma}_i(t)\sqrt{2\pi}} \exp\left(-\frac{(x_i(t+1) - \tilde{\mu}_i(t))^2}{2\tilde{\sigma}_i^2(t)}\right). \tag{9}$$

Maximizing the likelihood in Eq. (9) is equivalent to minimizing the negative log-likelihood using the estimated $\tilde{\mu}_i(t)$ and $\tilde{\sigma}_i^2(t)$, i.e.,

$$-\log\left(p\left(x_i(t+1); \tilde{\mu}_i(t), \tilde{\sigma}_i^2(t)\right)\right) = -\log\left(\frac{1}{\tilde{\sigma}_i(t)\sqrt{2\pi}}\right) \\ -\log\left(\exp\left(-\frac{(x_i(t+1) - \tilde{\mu}_i(t))^2}{2\tilde{\sigma}_i^2(t)}\right)\right) \tag{10}$$

Here, the constant coefficients and terms can be omitted, hence the loss function becomes

$$\mathcal{L}_i(\boldsymbol{\theta}_f, \boldsymbol{\theta}_g, \boldsymbol{\theta}_\phi | x_i(t+1), x_i(t), \mathrm{d}t) := \frac{(x_i(t+1) - \tilde{\mu}_i(t))}{\tilde{\sigma}_i^2(t)} + \log|\tilde{\sigma}_i^2(t)|. \tag{11}$$

For a training dataset containing $n$ observed nodes, the expectation becomes

$$\mathrm{argmax}_{\boldsymbol{\theta}_f, \boldsymbol{\theta}_g, \boldsymbol{\theta}_\phi} \frac{1}{n} \sum_{i=1}^{n} \log p_{\boldsymbol{\theta}_f, \boldsymbol{\theta}_g, \boldsymbol{\theta}_\phi}(x_i(t+1)|x_i(t), \mathrm{d}t). \tag{12}$$

For the case $d > 1$, the negative logarithm of a multivariate normal distribution can be written as

$$-\log(p(\mathbf{x}_i(t+1); \boldsymbol{\mu}(t), \Sigma(t))) = \frac{d}{2}\log(2\pi) + \frac{1}{2}\log|\Sigma(t)| \\ + \frac{1}{2}(\mathbf{x}_i(t+1) - \boldsymbol{\mu}(t))^T \Sigma(t)^{-1}(\mathbf{x}_i(t+1) - \boldsymbol{\mu}(t)), \tag{13}$$

where $\Sigma(t)$ is a positive semidefinite matrix.

## Inference of self, interaction, and diffusion parts

After the well-trained model of Fig. 1b separates the self, interaction, and diffusion parts, we adopt the core idea of the two-phase inference approach proposed by us previously[17] to infer the concise form for each part. Specifically, with three pre-constructed extensive libraries $L_F$, $L_G$, and $L_\Phi$ that contain widely used elementary functions (see Supplementary Information Section I-B), we introduce the time series data $\mathbf{x}_i(t)$, where $i \in n$, into $L_F$, $L_G$, and $L_\Phi$, and obtain the time-varying matrices $\Theta_F(t) \equiv L_F(\mathbf{x}_i(t))$, $\Theta_G(t) \equiv L_G(\mathbf{x}_i(t), \mathbf{x}_j(t))$, and $\Theta_\Phi(t) \equiv L_\Phi(\mathbf{x}_i(t))$. Then, the inference problem can be formulated using the estimated values as follows:

$$\begin{aligned} \hat{\mathbf{f}}(\mathbf{x}_i(t)) &= \tilde{\Theta}_F(t)\boldsymbol{\xi}_F, \\ \hat{\mathbf{g}}(\mathbf{x}_i(t), \mathbf{x}_j(t)) &= \tilde{\Theta}_G(t)\boldsymbol{\xi}_G, \\ \hat{\boldsymbol{\phi}}(\mathbf{x}_i(t))/\sqrt{\mathrm{d}t} &= \tilde{\Theta}_\Phi(t)\boldsymbol{\xi}_\Phi. \end{aligned} \tag{14}$$

Here $\tilde{\Theta}_F \equiv \Theta_F \otimes I_d$, $\tilde{\Theta}_G \equiv \Theta_G \otimes I_d$, and $\tilde{\Theta}_\Phi \equiv \Theta_\Phi \otimes I_d$, where $\otimes$ is the Kronecker product, and $I_d$ is $d \times d$ identity matrix. Therefore, the objective is to find the appropriate sparse coefficients $\boldsymbol{\xi}_F, \boldsymbol{\xi}_G$, and $\boldsymbol{\xi}_\Phi$, where most of the elements are zero while only the coefficients of highly relevant elementary functions are non-zero, such that Eq. (14) closely matches the observed data.

The first phase involves global regression to find a few most relevant elementary functions for each part, based on the optimization formulas:

$$\mathrm{argmin}_{\boldsymbol{\xi}_F} \int_0^T \sum_{i=1}^{n} \left(||\tilde{\Theta}_F(t)\boldsymbol{\xi}_F - \hat{\mathbf{f}}(\mathbf{x}_i(t))||^2\right)\mathrm{d}t + \lambda_F(||\boldsymbol{\xi}_F||_1)$$

$$\mathrm{argmin}_{\boldsymbol{\xi}_G} \int_0^T \sum_{i=1}^{n} \sum_{j=1}^{n} \left(||\tilde{\Theta}_G(t)\boldsymbol{\xi}_G - \hat{\mathbf{g}}(\mathbf{x}_i(t), \mathbf{x}_j(t))||^2\right)\mathrm{d}t + \lambda_G(||\boldsymbol{\xi}_G||_1) \tag{15}$$

$$\mathrm{argmin}_{\boldsymbol{\xi}_\phi} \int_0^T \sum_{i=1}^{n} \left(||\tilde{\Theta}_\phi(t)\boldsymbol{\xi}_\phi - \hat{\boldsymbol{\phi}}(\mathbf{x}_i(t))/\sqrt{\mathrm{d}t}||^2\right)\mathrm{d}t + \lambda_\phi(||\boldsymbol{\xi}_\phi||_1),$$

where $\lambda_F$, $\lambda_G$ and $\lambda_\phi$ are hyper-parameters that regulate the sparsity of the coefficients. In the implementation, we use the least absolute shrinkage (LASSO) with five-fold validation to determine the optimal hyper-parameters. Through global regression, we obtain the degree of relevance between each elementary function in the libraries and the hidden dynamics, significantly reducing the model space.

Next, we utilize the second phase to identify the minimal number of elementary functions for self, interaction, and diffusion parts, respectively, which constitute the final stochastic differential equation. To do so, we add the most relevant elementary functions one by one according to the relevance degree obtained in the first phase. We use metric $\kappa^2 = 1 - \frac{\sum_i (\hat{y}_i - y_i)^2}{\sum_i (y_i - \bar{y})^2}$ to indicate the regression score of a temporary combination of elementary functions. The more accurate is the current equation, the closer to 1 is $\kappa^2$. Here, $\hat{y}$, $y_i$, and $\bar{y}_i$ are prediction, true value and mean of true value, respectively. As we sequentially add the relevant elementary functions into equation, the metric $\kappa^2$ will change accordingly. The minimal number of elementary functions for each part is determined when adding more elementary functions into the equation does not increase, or even decreases, the value of $\kappa^2$, as shown in Fig. 1e–g.

## Quantification of inference inaccuracy

The goal of our study is to infer the mathematical equation that describes the dynamics underlying a complex system rather than only to predict future states of the system. To quantify the difference between the two equations, we use symmetric mean absolute

percentage error (sMAPE):

$$\text{sMAPE} = \frac{1}{k}\sum_{i=1}^{k}\frac{|I_i - R_i|}{|I_i| + |R_i|}. \tag{16}$$

Here, $k$ is the cardinal number of the set containing the inferred and true terms, $I_i$ and $R_i$ are the inferred and true coefficients for each term respectively. The value of sMAPE is within the interval between 0 and 1. The smaller the sMAPE, more accurate is the inference result. Importantly, sMAPE is sensitive to false negative and false positive errors. For example, if the inferred equation contains a term that should not be there, or does not contain a term that should be there, the value of sMAPE will increase significantly.

## Reporting summary
Further information on research design is available in the Nature Portfolio Reporting Summary linked to this article.

## Data availability
The codes for generating the simulation data in this study are deposited in the public GitHub repository[51]. The real data of bird flocks[40] and tau pathology[49] are shared via the link https://doi.org/10.6084/m9.figshare.24804894.v4. Source data are provided in this paper.

## Code availability
The codes are available in the public GitHub repository and on Zenodo: https://github.com/Ting-TingGao/Network-SDE-Inference.git[https://doi.org/10.5281/zenodo.12112887][51].

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

## Acknowledgements

GY is supported by the National Natural Science Foundation of China (grants no. T2225022, no. 12161141016, no. 12350710786, and no. 62088101), STI2030 Major Project (grant no. 2021ZD0204500), Shanghai Municipal Science and Technology Major Project (grant no. 2021SHZDZX0100), Shuguang Program of Shanghai Education Development Foundation and Shanghai Municipal Education Commission (grant no. 22SG21), and the Fundamental Research Funds for the Central Universities. BB is supported by the Israel Science Foundation (grant no. 499/19), the Israel-China ISF-NSFC joint research program (grant no. 3552/21), the US National Science Foundation CRISP award no. 1735505, and the VATAT grant for data science research. The authors are also grateful for the helpful discussion with Zhuohao He, Jack M. Moore, Xiaozhu Zhang, Tongyu Li, and Xiaolei Ru.

## Author contributions

G.Y. conceived the research, G.Y., T.T.G. and B.B. designed it, T.T.G. performed it, G.Y., T.T.G., and B.B. analyzed the results, and G.Y. and T.T.G. wrote the manuscript with input from B.B.

## Competing interests

The authors declare no competing interests.
