## [Peer Review File · Nature Communications]

REVIEWER COMMENTS

Reviewer #1 (Remarks to the Author):

In the revision, the authors have made significant changes to respond to the reviewers' comments. According to the clarification, it now becomes clearer that the manuscript provides an important method of inferring the SDE when given the network topology for the network dynamics. Therefore, this manuscript may be worth publishing in Nature Communications. Before recommending the publication, I would suggest the authors further address the following questions.

1. I agree with the authors that observational noise and intrinsic stochasticity are two different things. However, the difficulty is that for a real system, distinguishing whether the noise is intrinsic or extrinsic is challenging. This includes many cases in biology, for example, see the seminar work [Intrinsic and extrinsic contributions to stochasticity in gene expression]. There is a lot of literature on how to distinguish between these two sources of noise. The issue here is that, for a given set of real data, how do we know whether the noise is mainly from intrinsic or extrinsic sources? If there is no prior knowledge on that, I would regard treating the noise as intrinsic as an assumption. Then, the authors should make a clarification on this point about this assumption.

2. I appreciate the authors' efforts in detailing the differences with the previous methods. Now it is transparent that the present method outperforms the previous ones in inferring the networked stochastic Lorenz system. However, I wonder whether this is also the case when the data is not generated from predefined network dynamics, but from a more general SDE that goes beyond a predefined pairwise linear interaction term. For example, how about just the original 3-dimensional Lorenz system added with white noise?

Besides, since the major goal of all these inference methods is to deal with real data, a more convincing comparison would be to test these methods on some real data, such as the AD dataset in this manuscript. At least some discussions should be included on the scope of the comparison and the potentially different performances of these methods for real data.

Reviewer #1 (Remarks on code availability):

As mentioned in my previous review, the code is well organized with an informative Readme file. Now, the authors have also added more documentation to the code.

Reviewer #2 (Remarks to the Author):

Compared to the manuscript version in NMI, this updated manuscript has made several improvements that addresses several of my concerns. Overall, the strengths and limitations of the updated manuscript are as follows:

Strengths:

The paper addresses a very important problem, and the proposed method will be useful to a wide range of applications. The experimental validation is sound, especially after the added comparisons with several baselines.

Limitations:

The novelty, in my opinion, is limited, compared to prior works including the different prior graph neural network architectures and the authors' prior work Nat. Comput. Sci. 2, 160–168 (2022). Also, there is limitation that the method requires knowledge of the underlying topology.

In my opinion, the strengths slightly outweighs the limitations.

It would be great if the authors can answer my following two questions:

1. In the Figure 2m, why is the proposed method significantly outperforms the baselines (by two orders of magnitude)? What is the underlying mechanism behind it?
2. What is the mechanism that the authors use to set the threshold for setting the coefficients to be zero? In real applications, there are no ground-truth, a reliable way to set the threshold is important.

Response to Reviewers' Comments:

Learning interpretable dynamics of stochastic complex systems from experimental data

Ting-Ting Gao, Baruch Barzel, and Gang Yan.

Response to Reviewer #1

In the revision, the authors have made significant changes to respond to the reviewers' comments. According to the clarification, it now becomes clearer that the manuscript provides an important method of inferring the SDE when given the network topology for the network dynamics. Therefore, this manuscript may be worth publishing in Nature Communications. Before recommending the publication, I would suggest the authors further address the following questions.

We appreciate the positive feedback regarding the revision and the recognition of the achieved clarity. Your three constructive comments, which we addressed in the revised version, have helped us further improve the quality of our work.

1. I agree with the authors that observational noise and intrinsic stochasticity are two different things. However, the difficulty is that for a real system, distinguishing whether the noise is intrinsic or extrinsic is challenging. This includes many cases in biology, for example, see the seminar work [Intrinsic and extrinsic contributions to stochasticity in gene expression]. There is a lot of literature on how to distinguish between these two sources of noise. The issue here is that, for a given set of real data, how do we know whether the noise is mainly from intrinsic or extrinsic sources? If there is no prior knowledge on that, I would regard treating the noise as intrinsic as an assumption. Then, the authors should make a clarification on this point about this assumption.

Thank you for this insightful and constructive comment. We agree that observational noises also exist in real system data, and distinguishing between intrinsic and extrinsic noises can pose challenges. In cases where no prior knowledge is available regarding the dominant source of noise in a given set of real data, we acknowledge that treating all noises as intrinsic is an assumption we made in the present work.

If the observational noise is strong, we have followed the reviewer's suggestion by incorporating a preprocessing step to distinguish between intrinsic and extrinsic noises. Illustrated in Fig. R1, we have taken the networked Lorenz system with an intrinsic stochasticity intensity of $1/\sqrt{\gamma} = 1$ as an example. We introduced observational noise $\sigma\nu(t)$ into the data, where σ represents the standard deviation of the original stochastic time series, $\nu(t)$ is a random number drawn from a

Gaussian distribution with a mean of zero and a variance of one for time step t , and $p \in [0, 1]$ denotes the relative strength of observational noise. The outcomes revealed that when treating all noises as intrinsic, our method LaGNA demonstrates accurate inference of the stochastic dynamics of the networked system when the relative strength of observational noise is below 10%. However, when the observational noises are more pronounced, the preprocessing step of distinguishing between intrinsic and extrinsic noises (in this case, we used the Kalman-Takens filter) proves beneficial in enhancing the capability of our method (refer to Fig. R1(b)). This enhancement becomes particularly notable when inferring the stochastic dynamics of single-node systems (as depicted in Fig. R2 below).

In the revised version, we have discussed this point in lines 260 to 266 at page 8, and added the results in SI Sec. V-B.

Fig. R1. (a) Preprocessing by Kalman-Takens filter. (b) Inference inaccuracy (sMAPE) of our method LaGNA with or without the preprocess of Kalman-Takens filter versus relative strength of observational noise.

2. I appreciate the authors' efforts in detailing the differences with the previous methods. Now it is transparent that the present method outperforms the previous ones in inferring the networked stochastic Lorenz system. However, I wonder whether this is also the case when the data is not generated from predefined network dynamics, but from a more general SDE that goes beyond a predefined pairwise linear interaction term. For example, how about just the original 3-dimensional Lorenz system added with white noise?

To address this legitimate comment, we conducted a comparison study on the single-node stochastic system as you suggested. Similar to the above, we introduced observation noise $\sigma\nu(t)$ to the single-node Lorenz system with an intrinsic stochasticity intensity of $1/\sqrt{\gamma} = 1$. As illustrated in Fig. R2, our method LaGNA is capable of inferring the SDE of system dynamics when the observation noise is at the level of 10%, and it outperforms the five existing methods. Moreover, for both levels (10% and 50%) of observational noises, the preprocessing step of the Kalman-Takens filter consistently enhances the capability of our method LaGNA.

In the revised version, we have added this new result into SI Sec. V-C.

Fig. R2. Inference inaccuracy (sMAPE) of different methods in inferring the SDE of single-node Lorenz system when the relative strength of observational noise is 10% (a) or 50% (b). The bar chart depicts the sMAPE of five independent runs, along with the standard deviation indicated.

3. Besides, since the major goal of all these inference methods is to deal with real data, a more convincing comparison would be to test these methods on some real data, such as the AD dataset in this manuscript. At least some discussions should be included on the scope of the comparison and the potentially different performances of these methods for real data.

Following your suggestion, we conducted an additional comparative study on the AD propagation dataset. Specifically, we implemented three existing methods for inferring stochastic systems: SFI, SDE-net, and SVISE. To ensure a fair comparison, we extended these methods to apply to complex networked systems. Using the inferred equations from these methods, we generated propagation trajectories based on the initial state of injected regions. The correlation between the predicted and the true pathologies is shown in Fig. R3. The results demonstrate that the equations inferred by these three existing methods have significantly less predictive power compared to our method (Fig. 4b in the main paper).

In the revised version, we have added this additional comparison into SI Sec. V-D.

Remarks on code availability:

As mentioned in my previous review, the code is well organized with an informative Readme file. Now, the authors have also added more documentation to the code.

Thanks.

Fig R3. The predictions of tau pathology from the equations inferred by SFI, SDE-net, and SVISE, compared to the true tau pathology at 1, 3, 6, and 9 MPI respectively. The square of Pearson correlation R^2 for multiple linear regression tests are noted.

Response to Reviewer #2

Compared to the manuscript version in NMI, this updated manuscript has made several improvements that addresses several of my concerns. Overall, the strengths and limitations of the updated manuscript are as follows:

Strengths:

The paper addresses a very important problem, and the proposed method will be useful to a wide range of applications. The experimental validation is sound, especially after the added comparisons with several baselines.

We are delighted that you recognized the importance and wide applicability.

Limitations:

The novelty, in my opinion, is limited, compared to prior works including the different prior graph neural network architectures and the authors' prior work Nat. Comput. Sci. 2, 160–168

(2022). Also, there is limitation that the method requires knowledge of the underlying topology.

In my opinion, the strengths slightly outweighs the limitations.

Thanks. We objectively discussed the limitations of our approach and carefully addressed your following two questions in this revision.

It would be great if the authors can answer my following two questions:

1. In the Figure 2m, why is the proposed method significantly outperforms the baselines (by two orders of magnitude)? What is the underlying mechanism behind it?

The previous methods, namely Modified-SINDy, Two-phase, SDEnet, SVISE, and SFI, can be categorized into two distinct types. The first two methods focus on inference of ordinary differential equations (ODEs) for single-node or networked systems, while the latter three methods specifically target the inference of stochastic differential equations (SDEs) for single-node systems. As comparisons were requested, we extended these baseline methods to be applicable to networked systems. Despite the extension, our LaGNA method still significantly outperforms these baselines for the two key reasons outlined below.

Fig. R4. The estimated drift and diffusion fields by methods SDEnet and SFI, and their comparisons with the true values. The node-state time series data are numerically generated using a networked stochastic system of Lorenz dynamics.

(1) Previous methods focused on separating the drift field (deterministic part) and the diffusion field (stochastic part) from time series data. For example, as shown in Fig. R6, the methods SDE-Net and SFI work well in separating drift and diffusion fields. However, in stochastic networked systems, the drift field contains not only the self-dynamics $\mathbf{F}(\mathbf{x}_i)$ of each node but also the interaction dynamics $\mathbf{G}(\mathbf{x}_i, \mathbf{x}_j)$ between nodes. The previous methods are not good at further separating these two dynamics. In contrast, LaGNA incorporates three neural-network-based modules, especially the message-passing module defined on links to capture the interaction dynamics, which together distinguish between self, interaction, and diffusion effects in observation

data. This distinction enables accurate learning of networked SDEs from stochastic trajectories, effectively overcoming the limitations of previous methods.

In the revised version, we have added this point in lines 132 to 136 at page 4.

(2) The inaccuracy significantly lower by two orders of magnitude is also attributed to the nature of sMAPE metric. The definition of sMAPE indicates that the occurrence of any incorrect selection (such as selecting non-existent terms or failing to select existing terms) results in a substantial increase of $1/n$ in error, where n is the number of terms in the inferred equation. Hence, if one or two terms are incorrectly identified, the inference inaccuracy sMAPE increases significantly (please see lines 348 to 351 at page 11 in the main paper).

2. What is the mechanism that the authors use to set the threshold for setting the coefficients to be zero? In real applications, there are no ground-truth, a reliable way to set the threshold is important.

Thank you for raising this important question, as it has enabled us to improve the clarity of the inference process.

Once our LaGNA model is well-trained, we proceed to penetrate the self, interaction, and diffusion modules respectively. The penetration procedure involves two steps: (1) a global regression to determine the minimal number k of required terms, and (2) a fine-tuning analysis to identify the specific terms. During these steps, we do not set a hard threshold but make decisions based on regression curves. Below, we use the AD propagation data as an example to detail the penetration procedure.

In step 1, we identify the top 5 terms with largest regression coefficients and sequentially add them to the equation, then calculate the regression score (R^2) and Akaike Information Criterion (AIC) of the equation with one to five terms respectively. For example, as shown in Figs. R5-left and R6-left, adding only one term (except the ‘constant’) with the largest coefficient already leads to a regression score larger than 0.9. Further addition of the remaining terms does not significantly increase the regression score or significantly decrease of the AIC. Hence, the number k of required term is determined to be one (except the ‘constant’).

In step 2, we assess each combination of k terms in the top 5 preliminary terms:

- If a single combination has a significantly higher regression score than other combinations, that combination of terms is finally selected for the equation. For example, in Fig. R5-right (where $k = 1$), among the four terms (*i.e.*, Exp, Tanh, Hill, Sigmoid), we find that the sigmoid term achieves the highest regression score and is therefore selected as the expression for the

retrograde interaction dynamics in the AD propagation data.

- If multiple combinations achieve similar high regression scores, we select the term with the lowest complexity. For example, in Fig. R6-right (where $k = 1$), the four terms achieve similar regression scores individually. Hence, we proceed to a simple step 3 to analyze the complexity. As shown in Fig. R6 (Step 3), each term can be represented by a binary tree, and the complexity of a mathematical expression is defined as $Complexity = D + N_{leaf}$, where D is the depth of the tree and N_{leaf} is the number of leaves in the tree. Since the Exp term has the lowest complexity, it is selected to capture the spatial dynamics in the AD propagation data.

After determining the necessary terms, we obtain the coefficient for each term by regressing the observed time series data. Note that the constant term is assessed separately. We found that adding a constant to the interaction dynamics of the inferred equation does not significantly increase the final regression score for AD propagation. Consequently, the inferred equation (Eq. 5 in the main paper) does not include a constant term in the retrograde, anterograde, or spatial propagation parts.

In summary, during the penetration procedure, we first determine the necessary terms and then obtain their coefficients. Since our objective is to infer the most concise SDE from the data, we prioritize having a minimal number of terms and those with the lowest complexity.

In the revised version, we have added the details of the penetration procedure in SI Sec. III-C.

Fig. R5. Identifying the terms to represent the retrograde interaction dynamics in AD propagation.

Step 1: Global regression

Step 2: Fine-tuning

Step 3: Complexity (Depth + Number of leaves) for each term

Fig. R6. Identifying the terms to represent the spatial interaction dynamics in AD propagation.

REVIEWERS' COMMENTS

Reviewer #1 (Remarks to the Author):

The authors have adequately addressed all my previous comments. They have added more detailed analysis about the noise filtering and comparisons with other methods, which leads to a better study of learning network dynamics from the data. Thus, I would like to recommend its publication to Nature Communications.

Reviewer #1 (Remarks on code availability):

The code is well documented.

Reviewer #2 (Remarks to the Author):

After reading the response, the authors adequately addressed my questions. I have no problem recommending the acceptance of the paper. I think that the paper is a solid contribution with potentially wide applications.

Response to Reviewers' Comments:

Learning interpretable dynamics of stochastic complex systems from experimental data

Ting-Ting Gao, Baruch Barzel, and Gang Yan.

Response to Reviewer #1

The authors have adequately addressed all my previous comments. They have added more detailed analysis about the noise filtering and comparisons with other methods, which leads to a better study of learning network dynamics from the data. Thus, I would like to recommend its publication to Nature Communications.

We thank the reviewer's constructive comments, and we are delighted that we have addressed all the comments.

Remarks on code availability:

The code is well documented.

Thank you.

Response to Reviewer #2

After reading the response, the authors adequately addressed my questions. I have no problem recommending the acceptance of the paper. I think that the paper is a solid contribution with potentially wide applications.

We are delighted that we have addressed all the comments. We appreciate the reviewer's constructive comments and the recognition of our contribution.